# The Link of mRNA and rRNA Transcription by PUF60/FIR through TFIIH/P62 as a Novel Therapeutic Target for Cancer

**DOI:** 10.3390/ijms242417341

**Published:** 2023-12-11

**Authors:** Kouichi Kitamura, Tyuji Hoshino, Atsushi Okabe, Masaki Fukuyo, Bahityar Rahmutulla, Nobuko Tanaka, Sohei Kobayashi, Tomoaki Tanaka, Takashi Shida, Mashiro Ueda, Toshinari Minamoto, Hisahiro Matsubara, Atsushi Kaneda, Hideshi Ishii, Kazuyuki Matsushita

**Affiliations:** 1Department of Laboratory Medicine, Chiba University Hospital, Chiba 260-8677, Japan; k.kitamura.0623@chiba-u.jp (K.K.); tananobu1229@yahoo.co.jp (N.T.); skobayashi@iuhw.ac.jp (S.K.); 2Department of Molecular Diagnosis, Graduate School of Medicine, Chiba University, Chiba 260-8670, Japan; tomoaki@restaff.chiba-u.jp; 3Department of Molecular Design, Graduate School of Pharmaceutical Sciences, Chiba University, Chiba 260-8675, Japan; hoshino@chiba-u.jp; 4Department of Molecular Oncology, Graduate School of Medicine, Chiba University, Chiba 260-8670, Japan; aokabe@chiba-u.jp (A.O.); fukuyo@chiba-u.jp (M.F.); bahityar@hotmail.com (B.R.); kaneda@chiba-u.jp (A.K.); 5Department of Medical Technology and Sciences, Health and Sciences, International University of Health and Welfare, Chiba 286-8686, Japan; 6Research Team for Promoting Independence and Mental Health, Tokyo Metropolitan Institute for Geriatrics and Gerontology, Tokyo 173-0015, Japan; t_shida@tmig.or.jp; 7Master’s Program in Medical Sciences, Graduate School of Comprehensive Human Sciences, University of Tsukuba, Tsukuba 305-8575, Japan; s2221355@s.tsukuba.ac.jp; 8Division of Translational and Clinical Oncology, Cancer Research Institute, Kanazawa University, Kanazawa 920-1192, Japan; minamoto@staff.kanazawa-u.ac.jp; 9Department of Frontier Surgery, Graduate School of Medicine, Chiba University, Chiba 260-8670, Japan; matsuhm@faculty.chiba-u.jp; 10Medical Data Science, Center of Medical Innovation and Translational Research (CoMIT), Osaka University, Osaka 565-0871, Japan; hishii@gesurg.med.osaka-u.ac.jp

**Keywords:** FUBP1-interacting repressor (FIR), aberrant RNA splicing, ribosomal RNA (rRNA), RPB6 of RNA polymerase (RNAP), transcription factor IIH (TFIIH) P62 PH (pleckstrin homology)

## Abstract

The interaction between mRNA and ribosomal RNA (rRNA) transcription in cancer remains unclear. RNAP I and II possess a common N-terminal tail (NTT), RNA polymerase subunit RPB6, which interacts with P62 of transcription factor (TF) IIH, and is a common target for the link between mRNA and rRNA transcription. The mRNAs and rRNAs affected by FUBP1-interacting repressor (FIR) were assessed via RNA sequencing and qRT-PCR analysis. An FIR, a c-myc transcriptional repressor, and its splicing form FIRΔexon2 were examined to interact with P62. Protein interaction was investigated via isothermal titration calorimetry measurements. FIR was found to contain a highly conserved region homologous to RPB6 that interacts with P62. FIRΔexon2 competed with FIR for P62 binding and coactivated transcription of mRNAs and rRNAs. Low-molecular-weight chemical compounds that bind to FIR and FIRΔexon2 were screened for cancer treatment. A low-molecular-weight chemical, BK697, which interacts with FIRΔexon2, inhibited tumor cell growth with rRNA suppression. In this study, a novel coactivation pathway for cancer-related mRNA and rRNA transcription through TFIIH/P62 by FIRΔexon2 was proposed. Direct evidence in X-ray crystallography is required in further studies to show the conformational difference between FIR and FIRΔexon2 that affects the P62–RBP6 interaction.

## 1. Introduction

The regulation of mRNA and ribosomal RNA (rRNA) transcription, ribosomal protein (RP) synthesis, and ribosome biosynthesis (RiBi) are necessary for cell survival [1]. In humans, three RNAPs are critical for synthesizing RNAs: RNAPI for rRNA, RNAPII for mRNA, and RNAPIII for tRNA. The rRNA is transcribed from ribosomal DNA (rDNA) or genes for rRNA in the nucleolus, and the RPs constituting the ribosome are generated. The pleckstrin homology (P62 PH) subunit of TFIIH interacts with RPB6, a common amino-terminal tail (NTT) of RNAPI, RNAPII, and RNAPIII [2,3]. Thus, RBP6 is a key molecule for integrating mRNA and rRNA transcription; however, the coupling mechanism of RNAPI and RNAPII in mRNA and rRNA transcription in cancer remains unclear. RNAPI in the nucleoli directly activates the transcription of genes for rRNA under normal conditions; however, RNAPII engages in rRNA and mRNA transcription under an atypical condition [4]. Ribosomes comprise large and small subunits (the 60S and 40S) forming a complex of proteins and RNAs [5]. The small 40S ribosomal subunit contains 1 rRNA (18S) and 33 RPs, whereas the large 60S subunit contains 3 rRNAs (28S, 5.8S, and 5S) and 47 RPs [6]. The short arms of the acrocentric chromosomes (13, 14, 15, 21, and 22) contain 47S rDNA, arranged as tandem repeat clusters [7]. In human diseases, the fusion between these acrocentric chromosomes results in Robertsonian translocations (RTs) that lose the short arms containing 47S rDNA [7]. In RTs, insufficient rRNA synthesis and impaired RiBi induce “ribosomopathies”, often accompanied by malignant tumors [7,8].

The far upstream element (FUSE) is a sequence needed for the proper expression of the human *c-myc* gene, and FUSE-binding protein (FUBP) 1, which is a single-strand (ss) nucleic acid (DNA/RNA)-binding protein, transcriptionally activates *c-myc* [9]. Yeast two-hybrid analysis showed that FUBP1 binds to a protein with transcriptional inhibitory activity, referred to as FUBP1-interacting repressor (FIR) [10]. FIR suppresses the P89 subunit of transcription factor (TF) IIH consisting of a seven-subunit core [9]. Three TFIIH subunits, namely, P62, P89, and cyclin H, were consistently co-immunoprecipitated with FIR [9]. FIR is an exon5-lacking splicing variant of poly(U)-binding splicing factor 60 (PUF60) [11] and suppresses the c-myc transcription by repressing the DNA helicase activity of P89/TFIIH [12]. In particular, FIRΔexon2, a dominant negative splicing form of FIR, is scarcely expressed in normal cells but expressed in cancer with *c-myc* activation [13,14,15]. In these scenarios, this study determined whether a FIR splicing form potentially coactivates mRNA and rRNA transcription in cancer.

The FIR family consists of splicing variant forms (FIR, PUF60, and FIRΔexon2) and contains three RNA recognition motifs (RRMs): RRM1, RRM2, and RRM3 (U2AF homology motif, UHM) [12,16]. Except for cancer, the intractable diseases with insufficient rRNA synthesis, the germline variants were detected in the RRM1 and RRM2 sites required for FIR dimerization, indicating that FIR dimerization is necessary for rRNA synthesis [16]. Exon5 of PUF60 is pivotal for the regulation of neurogenesis through a mutual alternative splicing switch from PUF60 to FIR by remodeling splicing cofactor interactions [17]. The present study demonstrates the significance of coupling between rRNA and mRNA transcription in cancer in terms of the FIR-TFIIH/P62 interaction. A common and novel nucleolar rRNA and mRNA transcription pathway is proposed by FIR/FIRΔexon2 in cancer as well as intractable diseases.

Furthermore, small molecular chemical compounds against FIR and FIRΔexon2 were screened [18] among 23,275 chemicals by the Natural Product Depository (NPDepo) Array at RIKEN [19,20,21]. Among them, BK697, which potentially interacts with FIRΔexon2 [18], significantly inhibited tumor cell growth with the suppression of RP expression as a therapeutic strategy. The regulation of rRNA expression by FIR and FIRΔexon2 would support the notion suggesting that human congenital “ribosomopathies” exhibit a paradoxical transition from early symptoms due to cellular hypo-proliferation to elevated cancer risk later in life [22] with a clue for the treatment target.

## 2. Results

### 2.1. FIR and FIRΔexon2 Engaged in the Transcription of Various Cancer-Related mRNAs

Autoantibodies to FIR and FIRΔexon2 have been detected in various cancers [23] and autoimmune diseases [24], indicating that FIR and FIRΔexon2 peptides were recognized by the hosts’ immune system. FIR suppressed *c-myc* transcription through P89/TFIIH [9]; thus, FIR and FIRΔexon2 potentially affected the transcription of other nuclear genes as P89/TFIIH is a common TF that interacts with RNPII [9]. This study indicated that FIR and FIRΔexon2 affected the mRNAs transcription of various cancer-related genes in the nucleus regulated by RNPII. Since *FIRΔexon2* mRNA and its protein expression were first identified in HeLa cells [13], RNA sequencing in this study found that the expression of *FIR/PUF60* of itself, *FTH1*, *MT1E*, and *MTRNR2L2* was significantly affected by FIR and FIRΔexon2 in HeLa cells (Figure 1a). To support the FIR dimerization, pGBT9-FIR (as bait) interacted with pGAD-GH-FIR (as prey) in the yeast two-hybrid analysis. Out of 35, 28 (80%) positive clones were FIR of itself at 30 mM 3-aminotriazole (Appendix A). Similarly, the expression of *SLC3A2*, *TUBA1B*, *TUBA1C*, *TUBB*, *FSTL1*, and *INSIG1* was affected by FIRΔexon2 (Figure 1b), suggesting that FIR and FIRΔexon2 engage in the transcription of various cancer-related mRNAs. These differentially expressed genes (DEGs) expression levels were visualized in heatmaps (Figure 1a,b). The genes affected by up- (FLAG-tagged vector transfection) and down- (siRNA) expressions of FIR (Figure 1a) or FIRΔexon2 (Figure 1b) were indicated. *SNHG1* modulates *SLC3A2* transcription, leads the binding of FIR to FUBP1, and increases *Myc* transcription [25]. Furthermore, the gene expression level (Figure 1a,b) was compared with the Gene Expression Omnibus (GEO) database for colorectal cancer (Figure 1c), hepatocellular carcinoma (Figure 1d), lung cancer (Figure 1e), and gastric cancer (Figure 1f). FIR and FIRΔexon2 have been shown to be engaged in various nuclear mRNA genes’ transcription that were related to cancer in the GEO database. For instance, *FTH1*, *MT1E*, *SLC3A2*, *TUBA1B*, *TUBA1C*, *TUBB*, *FSTL1*, and *INSIG1* gene expression were significantly affected under siFIR and siFIRΔexon2 condition in HCT116 cells (Figure 1g). The endogenous gene candidates were *FIR/PUF60*, *FTH1*, and *MT1E* whereas *MTRNR2L2* in FIR and *SLC3A2*, *TUBA1B*, *TUBA1C*, *TUBB*, *FSTL1*, and *INSIG1* in FIRΔexon2. Similarly, the relative gene expression under the overexpression of FIR- and FIRΔexon2-FLAG was indicated in a histogram (Figure 1h). These results support the previous studies suggesting the relationship between the FIR and *SLC3A2* [25]. Together, FIR and FIRΔexon2 potentially activate the mRNA transcription of cancer-associated genes through common TF(s) regulated by RNPII in the nucleus.

### 2.2. FIR and FIRΔexon2 Affected Transcription of Various rRNAs

This study investigated the novel target genes except for cancer-related mRNAs in the nucleus, such as nucleolar rRNA. Alternative splicing of variant forms of FIR, FIRΔexon2, activates *c-myc* in cancers [9], whereas germline mutations of FIR cause “ribosomopathy” in human disease [26,27,28,29]. c-Myc enhances RNAPI activation by binding the rDNA promoter [30] and regulating protein synthesis via stimulation of RPs consisting of small ribosomal subunits (RPSs) and large ribosomal subunits (RPLs) [31]. Hence, dynamic nucleolar rRNA expression was examined in terms of the involvement of FIR and FIRΔexon2 via comprehensive RNA sequencing. Concrete RP genes (Figure 2a,b) are summarized in Appendix A. In the case of FIR, 31 (Group (Gr) 1) + 16 (Gr 2) = 47 genes were affected by FIR-FLAG overexpression, whereas 16 (Gr 2) + 3 (Gr 3) = 19 genes were affected by FIR knockdown by siRNA in HeLa cells (Appendix A). Similarly, in the case of FIRΔexon2, 22 (Gr 4) + 35 (Gr 5) = 57 genes were affected by FIRΔexon2-FLAG overexpression, whereas 35 (Gr 5) + 8 (Gr 6) = 43 genes were affected by FIRΔexon2 knockdown by siRNA (Appendix A) (Figure 2a,b). Several RPs and splicing factors were previously found to be co-immunoprecipitated with FIR-FLAG or FIRΔexon2-FLAG [15,30]. Briefly, the proteins that co-immunoprecipitated with FIR or FIRΔexon2 were identified via direct nanoflow liquid chromatography–tandem mass spectrometry analysis in 293T cells or via GeLC-MS of flag-conjugated bead pull-down with LC-MS in HeLa cells [30]. These results support the previous studies that both FIR and FIRΔexon2 participated in the posttranscriptional or translational processes [30].

In support of these findings, numerous rRNA-processing proteins were significantly affected by the upregulation (Figure 2c) and downregulation (Figure 2d) of FIR or FIRΔexon2 (Appendix A). Because FIRΔexon2 is detected in colorectal cancer tissues, the expressions of *RPL30*, *RPL37A*, *RPL38*, *RPS14*, and *RPS29* mRNAs by the effect of FIRΔexon2 were examined in HCT116 cells. Some of these RP mRNAs significantly decreased due to the overexpression of both FIR-FLAG and FIRΔexon2-FLAG. Similarly, the rRNA expressions of *RPL19*, *RPL37*, *RPS6*, *RPS10*, *RPS15A*, and *RPS21* significantly decreased due to the overexpression of FIRΔexon2-FLAG in HCT116 cells (Appendix A). The knockdown or overexpression of FIRΔexon2 by siRNA noticeably decreased the rRNA expressions of *RPL19*, *RPS6*, and *RPS10* (Appendix A). Furthermore, the rRNA expressions of *RPL37*, *RPS10*, *RPS15A*, *RPL6*, *RPL29*, and *RPL23A* were increased by the knockdown of both FIR and FIRΔexon2. The relative *RPS21* and *RPS29* mRNA expressions were significantly decreased by the overexpression of both FIR and FIRΔexon2-FLAG in HeLa cells (top, Appendix A). Contrarily, the relative *RPS15A* mRNA expression was noticeably decreased by siFIRΔexon2 in HeLa cells (bottom, Appendix A). The knockdown of FIR or FIRΔexon2 by siRNA affected the mRNA expressions of *RPS15A*, *RPS21*, and *RPS29* in HepG2 cells (Appendix A). In addition, the knockdown of FIR by siRNA affected the protein, and mRNA expressions of *RPS15A*, *RPS21*, and *RPS29* in T98G cells (Appendix A). Numerous RPs were co-immunoprecipitated with FIR and FIRΔexon2 [15,18,32]. This suggested that FIR and FIRΔexon2 coexisted and monitored the RP expression, particularly *RPS15A*, *RPS21*, and *RPS29*, and feedback on the dynamism of rRNA expression.

The RPs that constituted the ribosomes were generated from the rRNAs transcribed from rDNA by RNAPI in the nucleolus. In the case of FIR participation in dynamic rRNA transcription, effective accessibility of FIR to common transcriptional factors on rDNA is important. The rDNAs affected by the overexpression or knockdown of both FIR (Appendix A) and FIRΔexon2 (Appendix A) were located on chromosomes 17, 12, 19, 5, 2, 6, 9, and 1 (Appendix A). These results indicated that FIR and FIRΔexon2 targeted nucleolar rRNA genes on specific chromosomes other than acrocentric chromosomes 13–15, 21, and 22. Collectively, these results indicated that RNAPII directly operated on rRNA genes [6] in the human nucleoli, and FIR and FIRΔexon2 potentially affected nucleolar rRNA expression at least partly through RNAPII.

### 2.3. The Majority of rRNA Expressions by FIR and FIRΔexon2 Were Independent of c-Myc Activation

This study indicated that the rRNA expressions affected by FIR and FIRΔexon2 were at least partially independent of c-Myc activation. c-Myc enhances RNAPI activation by binding the rDNA promoter [31] and regulating protein synthesis via stimulation of RPs consisting of small ribosomal subunits (RPSs) and large ribosomal subunits (RPLs) [32]. RPs are associated with RiBi, cell proliferation, cell differentiation, and alternative splicing [33,34]. In cancer, rRNA expression is upregulated, and the production of RPs altering apoptosis inhibition and cell cycle arrest is induced [35,36,37,38,39]. Because FIRΔexon2 activates c-Myc [14], the present study determined whether FIR and FIRΔexon2 affect rRNA expression through c-Myc. In Figure 3a, Venn diagrams have shown the overlap of 10 c-Myc-related RP genes affected by the knockdown expression of FIR and FIRΔexon2 by siRNA in HeLa cells. The genes affected by both the up- and down−expressions of FIR were *RPL26*, *RPS27*, and *RPL39* (Figure 2a (Gr3) and Figure 3a). The eight rDNAs affected by both the up- and down-expressions of FIRΔexon2 were *RPL11*, *RPL3*, *RPL39*, *RPL41*, *RPL7*, *RPL7A*, *RPS25*, and *RPS3A* (Figure 2b (Gr6) and Figure 3a). *RPL39* was affected by the knockdown of both FIR and FIRΔexon2 by siRNA (Figure 3a,c). The rDNAs, *RPL26*, *RPS27*, and *RPL39,* were affected by both the increased and decreased expressions of FIR (Appendix A) (Figure 2a (Gr3)). The c-Myc expression was upregulated by the knockdown of FIR by siRNA (Figure 3b). Here, the c-Myc expression was evaluated using the average of fragments per kilobase of exon per million mapped sequence reads (FPKM) [40,41] in RPL or RPS genes that were commonly detected in duplicated experiments by the FIR or FIRΔexon2 knockdown by siRNA in HeLa cells under a cut-off level FDR of <0.05. FPKM is used to calculate the relative abundance of the transcript of each rRNA gene [24,25]. In the case of FIR, three genes were c-Myc-related RP genes (Appendix A (Gr3)) (Figure 3c,d (yellow arrows)). In the case of FIRΔexon2, eight genes were c-Myc-related RP genes (Appendix A (Gr6)) (Figure 3c,d, green arrows). *RPL39* has duplicated arrows (Figure 3c). However, most of those rRNAs were independent of the c-Myc expression (Figure 3c,d); thus, many rRNAs were changed by the knockdown of FIR or FIRΔexon2 through a c-Myc-independent mechanism. These results indicated that FIR and FIRΔexon2 engaged in rRNA expression, at least partly, beside the c-Myc pathway (Figure 3c,d).

The location of the affected genes for rRNA on chromosomes is presented in Appendix A. Regarding FIR, the number of upregulated chromosomes was evaluated. Among the 66 affected genes for rRNA, 48 were on chromosomes 17, 19, 1, 12, 5, 16, 2, 6, 9, and 11, accounting for 72.7% (48/66) of the top 10 chromosomes (Appendix A). FIRΔexon2-affected RP genes were on chromosomes 17, 1, 12, 6, and 19, accounting for 70.0% (70/100) among those of the top 10 chromosomes (Appendix A). Overlapping of FIR and FIRΔexon2 was observed on chromosomes 17, 12, 19, and 1. Human rDNAs were mainly mapped to acrocentric chromosomes 13, 14, 15, 21, and 22 [7]. However, the findings of this study strongly suggested that FIR and FIRΔexon2 activated the RP genes or rDNAs of specific chromosomes other than five acrocentric chromosomes (Appendix A) (Appendix A). Further research is required to reveal how FIR and FIRΔexon2, engaged in the transcription of various nucleolar rRNAs and nuclear mRNAs on specific chromosomes.

### 2.4. FIR and FIRΔexon2 Have Highly Homologous Sequence with RPB6

Subsequently, the common mechanism investigated how FIR and FIRΔexon2 activate the genes of cell proliferation-related rRNAs and mRNAs. As FIR and FIRΔexon2 transcribe the genes for rRNAs and mRNAs independent of the c-Myc activation, a common transcriptional molecule of RNAPI/II needs to be activated. The RPB6 of RNAPI/II/II interacted with the PH domain of the P62 subunit of TFIIH [3], and FIR contained the highly conserved acidic string region (376-KKEKEEEELFPESERPEM-394) with RPB6 between RRM2 and RRM3/UHM (Figure 4a,b and Appendix A). The three alternatively spliced forms of PUF60, FIR, and FIRΔexon2 are presented in Figure 4a. RRM1 and RRM2 were crucial for FIR homodimerization [42]. RRM1 was bound to the c-myc promoter and RRM2 to a transcriptional activator [7,16,43]. Because of the unavailability of the crystal structure for the acidic string region, this region was flexible and easily changed its conformation (Figure 4c). Protein structures were searched from the Protein Data Bank (PDB: https://www.rcsb.org/ (accessed on 5 December 2023). Previously, partial domain structures of FIR, RRM1 (PDB ID:2QFJ) [42], RRM2 (PDB ID:2KXH) [16], and UHM (PDB ID:3DXB) [9] were elucidated via X-ray crystallography. The FIR protein bound ssDNA of FUSE as a dimer in which only the RRM1 and RRM2 domains of each subunit interacted with ssDNA (Appendix A) [16,43]. In the yeast two-hybrid analysis, FIR as a bait (pGBT9-FIR) interacted with the FIR as prey (pGAD-GH-FIR), indicating FIR forms a homodimer (Appendix A). RNAPI, RNAPII, and RNAPIII contained the common NTT in RPB6. These acidic strings were highly conserved among FIR, UVSSA, XPC, TFIIEα, TP53, and DP1 (Figure 4b). D (Asp) and E (Glu) were acidic amino acids that target the P62-binding site [3,17]. The assigning of protein structure to the electron map of X-ray analysis of the central regions of RRM1, RRM2, and RRM3/UHM was presented in the electron density map with the assigned atom geometry (Figure 4d—(a) RRM1, (b) RRM2, and (c) RRM3/UHM). The residue names were labeled at their side chains. The RRM1, RRM2, and RRM3/UHM domains fitted well to the electron density map (Figure 4d). Accordingly, X-ray crystal analysis elucidated the relative positions of these three domains. However, the structures of the gray regions have not been determined (Appendix A). Considering that FIR has the homologous sequence with RPB6 in glutamic acid (E)-rich acidic amino acid sequence along with the aromatic (F) and hydrophobic (L) amino acids, FIR is a candidate molecule that interacts with P62 of TFIIH (Figure 4b).

### 2.5. FIR, but Not FIRΔexon2, Directly Interacted with P62 of TFIIH

Direct molecular interaction between FIR or FIRΔexon2 and P62 of TFIIH was investigated by isothermal titration calorimetry (ITC) measurement. As a result, FIR, but not FIRΔexon2, directly interacted with P62 of TFIIH. Because the P62 domain of TFIIH interacted with both RNAPI and RNAPII via a common RPB6 subunit [3], rRNA transcription is potentially interfered with by the FIR–P62 interaction. In HepG2 cells [44], the knockdown of FIR expression by siRNA caused significant suppression of P62 but not P89 (Figure 5a). The effect of FIR and FIRΔexon2 on the T98G (Figure 5b) and HCT116 (Figure 5c, arrows) cells were investigated. The significant suppression of P62 by FIR siRNA, but not FIRΔexon2 siRNA, was determined (Figure 5d). Isothermal titration calorimetry (ITC) measurement was performed as described previously [18]. The suppression of P62 by FIR siRNA (Figure 5c (arrows),d) depicted the interaction between FIR and P62, supported by the titration curve of P62 of TFIIH with FIR vis ITC measurement (Figure 5e); however, P62 did not interact with FIRΔexon2 (Figure 5f). These results suggested that FIR was required for the stable expression of P62, which was suggestive of molecular interaction between P62 and FIR (Figure 5e). Exothermicity peaked in the earlier injections and decreased in the subsequent injections. The dissociation constant was calculated as association constant Ka = 2.34 × 10^7^ M^−1^ from the ITC (Figure 5e). The affinity of the binding energy between P62 of TFIIH and FIR was enthalpically favorable as the ITC revealed that the interaction was exothermic. The upregulation (Appendix A) and downregulation (Appendix A) of FIRΔexon2 affected the transcription of different rRNAs and mRNAs. The P62-binding site existed between RRM2 and RRM3/UHM (U2AF homology motif) (Appendix A). These results indicated that FIR interacted with P62 of TFIIH. FIR rather than FIRΔexon2 affects nucleolar rRNA transcription in differentiated cells (Figure 5a–d).

### 2.6. Cell Growth Inhibition with Altered Expression of rRNA by a Low-Molecular-Weight Chemical BK697 against FIRΔexon2

Since FIRΔexon2 inhibits FIR-TFIIH/P62 interaction presumably as a dominant negative effect that is pivotal for rRNA and mRNA expression. Therapeutic application by a low-molecular-weight chemical BK697 against FIRΔexon2 was investigated. The dynamic stimulation of rRNA by FIR provided a novel insight into newer therapeutic targets for cancer treatment. Screening was performed on the computer to find synthesized chemicals mimicking the structure of the identified natural chemical compound A01 (1,4a-dimethyl-2,3,4,4a,9,9a-hexahydro-1H-fluorene-1,9-dicarboxylic acid) (Appendix A) by the Namiki database (Namiki Shoji Co., Ltd., Tokyo, Japan, https://www.namiki-s.co.jp/english/ (accessed on 5 December 2023) [18]. The binding affinity between FIR∆exon2 and BK697 was measured using the MicroCal VP-ITC equipment, indicating that BK697 interacted with FIR∆exon2 (Figure 6a). The titration curve of the TFIIH P62 PH domain with FIR∆exon2 suggested the presence of a molecular interaction between the two proteins. Exothermicity peaked in earlier injections and decreased in the subsequent injections. The association constant was calculated to be 1.9 × 10^7^ M^−1^ from the ITC. The binding of the TFIIH P62 PH domain and FIR wild was enthalpically favorable as the ITC measurement indicated that the interaction was exothermic (Figure 5e). The expressions of FIR and FIRΔexon2 were suppressed by BK697 in HeLa cells (Figure 6b). Because FIRΔexon2 was detected in HepG2 [44], BK697 effectively suppressed HepG2 cell growth with an IC_50_ of 17.8 µM (Figure 6c). In addition, BK697 suppressed the expressions of FIR and FIRΔexon2 mRNA levels in HepG2 cells (Figure 6d). The expressions of *RPS6*, *RPS10*, *RPS15A*, *RPL30*, and *RPL23A* were reduced by BK697 at an IC_50_ of 17.8 µM in HepG2 cells (Figure 6e). These results elucidated the novel mechanism indicating that FIRΔexon2 regulated dynamic rRNA expression in cancer. Explicitly, FIR binds to P62 in non-cancer conditions, and RNAPI/II was in a stable status (Figure. 7a). In this non-cancer condition, FIR represses *c-myc* transcription; accordingly, the transcription of rRNA and mRNA are regulated as physiological status under low *c-myc* expression [30,31]. In cancer, FIRΔexon2 inhibits the FIR–P62 interaction as dominant negative for FIR, triggering P62 shifts to interact with RPB6 of RNAPI/II rather than P62-FIR interaction for activating rRNA and mRNA activation (Figure. 7b). In other words, FIR suppresses rRNA and mRNA transcription via inhibiting P62 to interact with RBP6/RNAPI/II in normal cell non-cancer condition (Figure. 7a). Nevertheless, FIRΔexon2 inhibits FIR to access to P62 and promotes P62 and RPB6 of RNAPI/II interaction for rRNA and mRNA transcription in cancer (Figure 7b). Therefore, BK697 that targeted FIRΔexon2 could be used as a potential anticancer drug by suppressing rRNA and mRNA transcription via promoting FIR-P62 interaction. This study indicated the *c-myc*-independent mechanism by FIRΔexon2 contributes at least partly to the upregulation of rRNA and mRNA transcription in cancer cells (Figure 7b).

## 3. Discussion

This study demonstrated the novel coactivation mechanism of mRNA and rRNA underlying the role of FIR and its splicing variant, FIRΔexon2. Because FIR is immunoprecipitated with P89 and P62 of TFIIH [7], FIR potentially affects RNAPII-mediated transcription via TFIIH. As expected, FIR affected the mRNA expression of various cancer-related genes including FIR mRNA transcription, as revealed by the RNA-sequencing analysis (Figure 1a–f). Essentially, numerous rRNA-processing proteins were significantly changed by the up and downregulations of FIR and FIRΔexon2 (Appendix A) (Figure 2c,d). Accordingly, FIR and FIRΔexon2 affected the rRNA expressions of *RPL19*, *RPL23A*, *RPL30*, *RPL36*, *RPL37*, *RPL37A*, *RPL38*, *RPS6*, *RPS10*, *RPS14*, *RPS15A*, *RPS16*, *RPS19*, *RPS21*, and *RPS29* in HCT116 cells determined via RT-PCR (Figure 3a,b). These results indicated that FIR and FIRΔexon2 engage in the mRNA transcription of several nuclear genes (Figure 1a,b,g,h) and nucleolar rRNAs (Appendix A) (Figure 2 and Figure 3) through a common transcriptional pathway. Although c-Myc stimulates transcription of rRNA genes by RNAPI [31], the alterations of many rRNAs by FIR or FIRΔexon2 were independent of c-Myc activation (Figure 3c,d). Some rRNAs were accompanied by c-Myc activation (Figure 4c,d, with arrows) [31]; however, the expressions of many rRNAs were independent of c-Myc (Figure 4c,d, without arrows). FIR has a homologous sequence with RPB6 (Figure 4b), a common NTT of RNAPI and RNAPII, which potentially interacts with P62 of TFIIH [2,3]. Furthermore, FIR, but not FIRΔexon2, interacted with P62/TFIIH (Figure 5e,f), indicating a novel coactivation pathway of mRNA and rRNA transcription through FIR and FIRΔexon2. RNAPI and RNAPII interact with TFIIH/P62 via RBP6, which is a common target for the transcription of both mRNA and rRNA (Figure 7a) [3]. The relative *RPS21* and *RPS29* expressions were significantly decreased by the overexpression of both FIR and FIRΔexon2-FLAG, whereas siFIRΔexon2 significantly decreased the relative *RPS15A* expression in HeLa cells (Appendix A). RNAPII in the human nucleoli directly operated rRNA genes in impaired cellular status [4]. FIR and FIRΔexon2 affected nucleolar rRNA expression by interacting with the nucleolar P62 of TFIIH. Due to aberrant splicing, FIRΔexon2 was scarcely expressed in normal cells but overexpressed in cancer cells [13,14,15]. Thus, the switching of the FIR to FIRΔexon2 expressions was responsible for the aberrant rRNA transcription in cancer. Together, FIR and FIRΔexon2 notably generated the rRNA expression of various RPs on chromosomes 17, 12, 19, and 1. This suggested that FIR and FIRΔexon2 stimulated nucleolar rDNA or genes for rRNA transcription on specific chromosomes aside from acrocentric chromosomes 13, 14, 15, 21, and 22 (Appendix A) (Appendix A). Therefore, the common regulatory pathway of mRNA and rRNA by FIR was speculated to directly interact with the P62 PH domain of TFIIH that binds to RPB6 [3] (Figure 6e). In non-cancer cells (general physiological condition), the mRNA and rRNA transcription is controlled by FIR-P62/TFIIH binding that competes with the P62–TFIIH-RBP6 interaction (Figure 7a). Because FIR formed a heterodimer with FIRΔexon2 [16], FIRΔexon2 possibly interfered with the FIR–P62 interaction and P62 caused an interaction with RPB6 of PNAPI/II to nucleolus-dependent aberrant rRNA transcription (Figure 7b). As RNAPII directly operated on rRNA genes [6] in the human nucleoli, FIR and FIRΔexon2 potentially affected nucleolar rRNA expression through RNAPII. Collectively, P62 of TFIIH was co-immunoprecipitated with FIR [7], and RNAPI/II interacted with TFIIH via RPB6 [3]; the access of P62 to TFIIH to RBP6 of both RNAPI and RNAPII enforces rRNA transcriptional activation in cancer. FIR and FIRΔexon2 comprised the highly conserved E-rich acidic region (376-KKEKEEEELFPESERPEM-394) between RRM2 and UHM (Figure 4b and Appendix A). The P62 subunit of TFIIH directly interacted with the RPB6 of RNAPI and RNAPII [3]. In addition, the siRNA of FIR significantly suppressed P62 expression in HepG2 (Figure 5a), T98G (Figure 5b), and HCT116 (Figure 5c,d) cells. The ITC revealed the direct interaction between the P62 of TFIIH and FIR but not FIRΔexon2 (Figure 5e,f). A low-molecular-weight chemical potentially interacting with FIRΔexon2 was identified in the NPDepo screening (A01 of Appendix A). BK697 was designed via in silico screening potentially binding FIRΔexon2. The BK697 interaction with FIR∆exon2 was detected using the MicroCal VP-ITC equipment (Figure 6a). BK689 significantly suppressed the FIRΔexon2 expression (Figure 6b) and inhibited tumor cell growth (Figure 6c) and several rRNA expressions (Figure 6e). BK697 and its derivatives could be potent candidates for anticancer drugs targeting FIRΔexon2. The dynamic stimulation of rRNA by FIR could provide a novel insight into the therapeutic targets for cancer treatment.

This RNAPI to RNAPII switching of ribosomal rRNA transcription by FIRΔexon2 led to atypical transcription of rRNA and mRNA (Figure 7b). Furthermore, FIR formed a homodimer and heterodimerized with FIRΔexon2 through RRM1 and RRM2 (Figure 4a, Appendix A). Due to the FIR-FIRΔexon2 interaction, FIRΔexon2 promotes P62 to interact with RPB6 via forming FIR-FIRΔexon2 heterodimer rather than P62-FIR interaction and potentially activates mRNA and rRNA transcription through P62-RBP6/RNAPI/II axis (Figure 7b). Hence, the FIR homodimer was potentially pivotal for interaction with P62. FIR contained a highly conserved acidic region, a similar sequence of the RPB6 subunit that interacted with P62 of TFIIH (Figure 4b). Moreover, regarding the interaction between the P62 PH domain and FIR, the role of the second hydrophobic residue was considered critical in the interaction and was not conserved in the FIR (Figure 4b). In addition, the interaction of the P62 PH domain and the small region containing the two conserved hydrophobic residues was generally dispensable in other systems (e.g., TFIIE, XPC) (Figure 4b). Regarding TFIIE and XPC, primary interactions with TFIIH were induced by other regions of TFIIE and XPC. These results indicated that FIR and FIRΔexon2 stimulated rRNA expression through RNAPII and potentially integrated the expression of RPs (Appendix A). FIR and FIRΔexon2 directly interacted with the splicing factor SAP155 (SF3B1) [15]. SF3B1 formed a complex with FIR [16] that was required for proper pre-mRNA splicing of the PUF60/FIR gene [15]. Hence, SF3B1 was partly responsible for the expressions of PUF60, FIR, and FIRΔexon2 in their feedback of PUF60/FIR/FIRΔexon2 splicing.

In human disease with “ribosomopathy”, some germline variants of the FIR/PUF60 gene, generating truncated protein lacking RRM1, RRM2, P62-binding site, or RRM3/UHM, have been reported in Verheij and CHARGE syndromes (Appendix A) (Appendix A). Verheij and CHARGE syndromes potentially had a common pathogenic pathway from the aspects of rRNA transcription. CHD7 functions as a regulator of both nucleoplasmic and nucleolar genes and causes rRNA disorder [45]. Impaired FIR dimerization is critical for the pathogenesis of CHARGE and Verheij syndromes. Impaired CHD7 observed in CHARGE syndrome affected the BRG1 expression and chromatin remodeling of rRNA genes [46,47,48]. In particular, FIRΔexon2 participated in the multistep posttranscriptional regulation of BRG1, affecting epithelial–mesenchymal transition and promoting tumor proliferation and gastric cancer cell invasion [18]. Furthermore, variants in some of the rDNAs caused Diamond–Blackfan anemia, which is characterized by bone marrow failure with rare cancer predisposition syndromes from hypo- to hyper-proliferation [22,49]. Together, the up and downregulations of rRNA transcription were closely associated with the pathogenesis of human diseases, including cancer and neurodegenerative disorders [22,33,34,35,36].

The experimental evidence indicated that FIR formed a homodimer complex via X-ray crystallography (Appendix A) [9,12], immunoprecipitation [15], and yeast two-hybrid analysis (Appendix A). In addition, FIR formed a complex with SF3B1 and engaged in RNA splicing [14,15]. This study hypothesized that FIR homodimerization by RRM1+RRM2 is critical for the FIR–P62 interaction and potentially competed with the P62–RPB6 interaction (Figure 7). Direct experimental evidence would be required to demonstrate the interaction between FIR dimerization and P62 and its contribution to rRNA and mRNA transcription in X-ray crystallography structure analysis.

Regulation of the rRNA expression by FIR and FIRΔexon2 should be demonstrated in congenital “ribosomopathies” exhibiting a paradoxical transition from early symptoms to increased cancer risk due to cellular hypo-proliferation observed in RTs and Diamond–Blackfan anemia later in life. The germline mutation of FIR in RRM1 + RRM2 of CHARGE or Verheij syndrome would provide insights into the role of FIR variants in inhibiting homodimerization and affecting rRNA and mRNA transcription. The relationship between the increase in disordered rRNA transcription and RiBi also needs to be investigated. Further study is warranted in this field. The transcription of various nuclear mRNAs and nucleolar rRNAs was regulated by FIRΔexon2 as a dominant negative functioning form of FIR inducing aberrant rRNA and mRNA transcription. A novel role of FIRΔexon2 facilitated nuclear mRNA and nucleolar rRNA transcription through the P62/PH domain bound to RPB6 proposed in the crystal structure analysis, comprehensive RNA sequencing, and qRT-PCR. This study clarified the novel mechanism demonstrating that FIR homodimerization is critical for the regulation of the expression of nucleolar rRNA and nuclear mRNA via TFIIH/RNAPII. The effect of FIRΔexon2 on the P62/RPB6/RNAPI/II axis link between cancer development and neural disease through RNA splicing was also investigated. In addition, the present study indicated a low-molecular-weight molecule BK697, targeting FIRΔexon2, was proposed for cancer treatment. Nucleolar rRNA and nuclear mRNA were transcribed through the RNAPI/II-RBP6/P62 axis. In neurodevelopmental disorders and cancers, P62 interacted with FIR homodimers but not with FIR/FIRΔexon2 heterodimers. BK697, a FIRΔexon2 inhibitory chemical, could salvage FIR homodimerization and was used as a potential therapeutic agent. This study provides a novel insight into human diseases by stimulating dynamic RiBi through aberrant splicing and giving a hint regarding the therapeutic strategy. The primary limitation of this study is the lack of direct and molecular evidence to demonstrate the mechanism of interaction between FIR dimerization and P62 and its contribution to rRNA and mRNA transcription.

## 4. Materials and Methods

### 4.1. Cell Lines, Plasmids, and Plasmids Transfection

HeLa (human cervical carcinoma: https://www.atcc.org/products/crm-ccl-2 (accessed on 5 December 2023)), HepG2 (human hepatocellular carcinoma: https://www.atcc.org/products/hb-8065 (accessed on 5 December 2023)), HCT116 (human colorectal carcinoma: https://www.atcc.org/products/ccl-247 (accessed on 5 December 2023)), and T98G (a fibroblast-like cell isolated from the brain of a glioblastoma multiforme: https://www.atcc.org/products/crl-1690 (accessed on 5 December 2023)) cells were purchased from ATCC (https://www.atcc.org/ (accessed on 5 December 2023)) and cultured in Iscove’s modified Dulbecco’s medium supplemented with 10% FBS and 1% penicillin–streptomycin (ThermoFisher Scientific Diagnostics, San Franscisco, CA, USA). Cells were cultured at 37 °C in 5% CO_2_. Plasmids have been described in a previous report [13]. Briefly, FIR cDNA was inserted into a p3xFLAG-CMV-14 vector (Sigma-Aldrich, Tokyo, Japan), and FIRΔexon2 cDNA was inserted into a pcDNA 3.1 plasmid (ThermoFisher Scientific Diagnostics, San Franscisco, CA, USA). Plasmids have been described in a previous report [13]. Briefly, FIR cDNA was inserted into a p3xFLAG-CMV-14 vector (Sigma-Aldrich, Tokyo, Japan), and FIRΔexon2 cDNA was inserted into a pcDNA 3.1 plasmid (ThermoFisher Scientific Diagnostics, San Franscisco, CA, USA). HeLa cells were transfected with a FLAG vector, FIR-FLAG, and FIRΔexon2-FLAG. The plasmid transfection was performed as reported previously [13,14]. Briefly, cells were cultured in Dulbecco’s modified Eagle’s medium (ThermoFisher Scientific Diagnostics, San Franscisco, CA, USA) with 10% fetal calf serum. The cells were transfected using Lipofectamine Plus or Lipofectamine 2000 (ThermoFisher Scientific Diagnostics, San Franscisco, CA, USA). Cell cultures routinely used either 1 × 10^5^ cells per well in 6-welled plates or 5 × 10^5^ cells per 10 cm in dishes overnight.

### 4.2. RNA Extraction and Quantitative Reverse Transcription (qRT)-PCR

Total RNA was extracted from HeLa cells using the MagNA pure compact RNA isolation kit (Roche Diagnostics, Indianapolis, IN, USA). The cDNA was synthesized using the first strand cDNA synthesis kit for reverse transcription (Roche Diagnostics, Indianapolis, IN, USA). The primer sets for qRT-PCR of *FIR*, *FIRΔexon2*, RPLs (*RPL6*, *RPL19*, *RPL23A*, *RPL29*, *RPL30*, *RPL37A*, *RPL37*, and *RPL38*), and RPSs (*RPS6*, *RPS10*, *RPS14*, *RPS15A*, *RPS21*, and *RPS29*) have been described in Appendix A. Additionally, a housekeeping hypoxanthine phosphoribosyltransferase 1 (*HPRT*) gene was used as an internal control (Appendix A).

### 4.3. RNA-Seq and Data Analysis

RNA-seq analysis was performed on HeLa cells transfected with FIR-FLAG, FIRΔexon2-FLAG, FLAG vector, or untransfected samples, with siFIR, siFIRΔexon2, or siGL2. The RNA-seq libraries were constructed from the total RNA samples with the TruSeq Stranded mRNA Sample Prep Kit (Illumina, Tokyo, Japan), as per the manufacturer’s protocol. The Illumina HiSeq1500 or NextSeq500 platforms were used for deep sequencing. The sequenced reads from RNA-seq data were aligned by using Hisat2. In addition, Cufflinks were used for transcript assembly. Gene expression levels were expressed as FPKM [40,41]. Differential expression analysis was conducted with the TCC software package (https://bioconductor.org/packages/release/bioc/html/TCC.html (accessed on 5 December 2023)) in R 3.4 (https://bioc.ism.ac.jp/packages/3.7/bioc/vignettes/TCC/inst/doc/TCC.pdf (accessed on 5 December 2023)). The upregulated or downregulated genes were analyzed using the DAVID Bioinformatics resources (https://david.ncifcrf.gov/ (accessed on 5 December 2023)).

### 4.4. Western Blotting

Cells were dissolved with 1:20 β-mercaptoethanol in a 2× sample buffer (4× Laemmli sample buffer (Bio-Rad Laboratories, Inc., Hercules, CA, USA) was diluted to 2× by 10% of β-mercaptoethanol with distilled water) and incubated at 100 °C for 5 min. The cell lysates were measured for protein concentration using the Bradford protein assay (Bio-Rad Laboratories, San Franscisco, CA, USA). Then, 10 μg of cell lysates were separated into 7.5% or 10–20% XV PANTERA gels (D.R.C. Co., Ltd., Tokyo, Japan) using SDS-PAGE. Before overnight incubation with primary antibodies at 4 °C, the membranes were blocked with 0.5% skimmed milk in phosphate-buffered saline (PBS) for 1 h at room temperature. This was followed by three 10 min washes with 0.1% Tween 20 in PBS. The membranes were incubated with commercial secondary antibodies for 1 h at room temperature, followed by three 15 min washes with 0.1% Tween 20 in PBS. The primary and secondary antibodies used in this study have been listed in Appendix A. Antigens were detected using Amersham ECL Western blotting detection reagents (GE Healthcare, Tokyo, Japan). The results of Western blot films used in this study have been indicated in Appendix A.

### 4.5. siRNAs for Specific Proteins Expression

Specific FIR, FIRΔexon2, and GL2 (firefly luciferase GL2 as a negative control, NIPPON Gene Co., Ltd., Tokyo, Japan) siRNA duplexes were purchased from Sigma-Aldrich, Tokyo, Japan. The internal control was β-actin, and the siRNA concentration of siGL2, siFIR-1, and siFIRΔeon2 was 25 pmol. GL2 was used as a negative control which did not exist in the human genome. The target sequences for FIR and FIRΔexon2 siRNA oligonucleotides have been indicated in Appendix A and performed as per the previous report [13,14]. Briefly, transient siRNA transfection was performed using Lipofectamine 2000 (ThermoFisher Scientific Diagnostics, San Franscisco, CA, USA) as per the manufacturer’s instructions. The transfected cells were cultured for 72 h at 37 °C in a CO_2_ incubator. Previous reports indicated that the siFIRΔexon2, a spliced variant of FIR, was selectively knocked down in cells while maintaining intact FIR [18,50].

### 4.6. X-ray Crystal Structure Analysis

Protein structures were searched from the Protein Data Bank (PDB: https://www.rcsb.org/ (accessed on 5 December 2023)). Partial domain structures of FIR, RRM1 (PDB ID: 2QFJ) [42], RRM2 (PDB ID:2KXH) [16], and RRM3/UHM (PDB ID:3DXB) [9], were previously elucidated by X-ray crystallography. The recombinant FIRΔexon2 protein was expressed as the whole body. The purified FIRΔexon2 at a concentration of 6.0 mg/mL was crystallized by a precipitant solution of 0.1 M Tris HCl. It had a pH of 7.0 and contained 2.2 M ammonium sulfate and 0.2 M lithium sulfate. The X-ray diffraction was obtained at the resolution of 1.4 Å. A structural analysis was carried out by replacing the molecules with crystal structures, 2QFJ and 3DXB, as templates.

### 4.7. Protein Expression and Purification of FIR, FIR∆exon2, and p62 PH Domain

Proteins of FIR, FIR∆exon2, and P62 PH domain were expressed and purified as reported previously [26]. Briefly, the cDNA of *FIR*, *FIR∆exon2*, and p62 PH domain was inserted into a pET-50b (+) DNA plasmid vector. An Escherichia coli strain, Rosetta (DE3) pLysS (competent cells), which was transformed with the pET-50b-FIR, -FIR∆exon2, or -p62 PH vector, was cultured in 1 L Luria-Bertani (LB) medium at 37 °C until the O.D. 600 value reached 0.6. The protein expression was induced by an addition of 0.2 mM isopropyl β-D-1 thiogalactopyranoside (IPTG). The medium was incubated for the following 12 h at 28 °C for FIR and FIR∆exon2 and at 20 °C for the p62 PH domain. A cell pellet was obtained by centrifugation of the cultured medium. The pellet was resuspended in a buffer of 50 mM Tris-HCl at pH 8.0 and 500 mM NaCl containing 10 mM imidazole and 1 mM phenylmethylsulfonyl fluoride. After disrupting the bacterial cell membrane by sonication, the protein was purified with a co-affinity column with a gradient rise of the imidazole concentration. The eluted fraction was dialyzed overnight against the buffer without imidazole. The Nus-tag was cleaved by human rhinovirus (HRV) 3C protease and the protein was purified with a Ni-affinity column to remove the cleaved Nus-tag and HRV 3C protease. The protein was finally purified by gel filtration with a running buffer of 10 mM Tris-HCl at pH 8.0 and 300 mM NaCl.

### 4.8. ITC Measurement

The binding affinity between FIR wild or FIR∆exon2 and TFIIH p62 PH domain was measured using an ITC technique in the MicroCal VP-ITC system (Malvern Analytical, Malvern, UK), as described previously [31]. The sample cells were filled with 1400 μL of 50 mM phosphate buffer, pH 7.2, containing 4 μM purified wild FIR or 14.5 μM FIR∆exon2. The binding affinity was measured at 30 °C. A solution of 50 mM phosphate buffer, pH 7.2, containing 44 or 145 μM TFIIH p62 PH domain was injected into the sample cells with a syringe for titration. The volume of each injection was 10 μL and the injection time was 20 s and 150 s. The titration was repeated 25 times.

### 4.9. Procedure of In Silico Screening of Small Molecular Chemical Compounds BK697 against FIRΔexon2

Small molecular chemical compounds against His-tagged FIR (His-FIR, 645 μg/mL) and -FIRΔexon2 (His-FIRΔexon2, 652 μg/mL) were screened among 23,275 chemicals of NPDepo at RIKEN, Saitama, Japan) [19,20,21]. In the process of searching for potent compounds, in silico screening was performed from the commercial chemical database. First, 1000 compounds were selected from the Namiki database that contains 5 million chemical entries, from the viewpoint of structural similarity to natural product that was identified to be bound to FIR or FIRΔexon2 in our previous work (https://www.hrjournal.net/article/view/2819/2462 (accessed on 5 December 2023)) [18].

### 4.10. MTS Assay (Cell Proliferation Assay)

The MTS assay was performed as described previously [18,44]. Briefly, one day before the chemical treatment, cells were cultured in a 100 µL medium in flat-bottomed 96-welled plates to make the cells reach 40–80% confluent during the chemical treatment. After 24 h incubation at 37 °C/5% CO_2_, cells were treated with chemicals, including BK697. After 24 h incubation at 37 °C, CellTiter 96^®^ AQueous One Solution Reagent (Promega Corporation, Madison, WI, USA) was added to each well as per the manufacturer’s instructions. Briefly, CellTiter 96^®^ AQueous One Solution Reagent was warmed up and then added to each well (20 µL/well). It was incubated for 1 h at 37 °C. Then, 10% SDS solution was added to each well (25 µL/well). The cell viability was determined by measuring the absorbance at 490 nm using a 550 Bio-Rad plate reader. All samples were tested in duplicate. Also, absorbencies were tested thrice. The same volumes of dimethyl sulfoxide and 3% H_2_O_2_ were used as negative and positive controls, respectively.

### 4.11. Yeast Two-Hybrid Analysis

The yeast two-hybrid assay was carried out as per the manufacturer’s instructions (MATCHMAKER Library Protocol. Clontech, Washington, DC, USA) [9]. Briefly, 1 × 10^7^ HeLa cell cDNA library clones were amplified and screened; pGBT9-FIR was used as a bait plasmid and pGAD-GH-FBP was used as a positive control. Positive clones were selected for the expression of HIS3, TRP1, and LEU2 markers in the presence of 30 mM 3-amonotriazole followed by a b-galactosidase assay. The true positives were DNA sequenced. Gal4-FBPC was used as a transcriptional activator plasmid [9].

### 4.12. Statistical Analyses

The expression of FIR and FIRΔexon2 mRNA was compared with that of FIR-FLAG, FIRΔexon2-FLAG, and FLAG vector. The expression of siFIR and siFIRΔexon2 was compared with that of siGL2 by assessing the untreated sample using Student’s *t*-tests. Statistical analyses were conducted using GraphPad Prism version 6.0 for Windows (GraphPad Software, https://www.graphpad.com/ (accessed on 5 December 2023)).

## Figures and Tables

**Figure 1 ijms-24-17341-f001:**
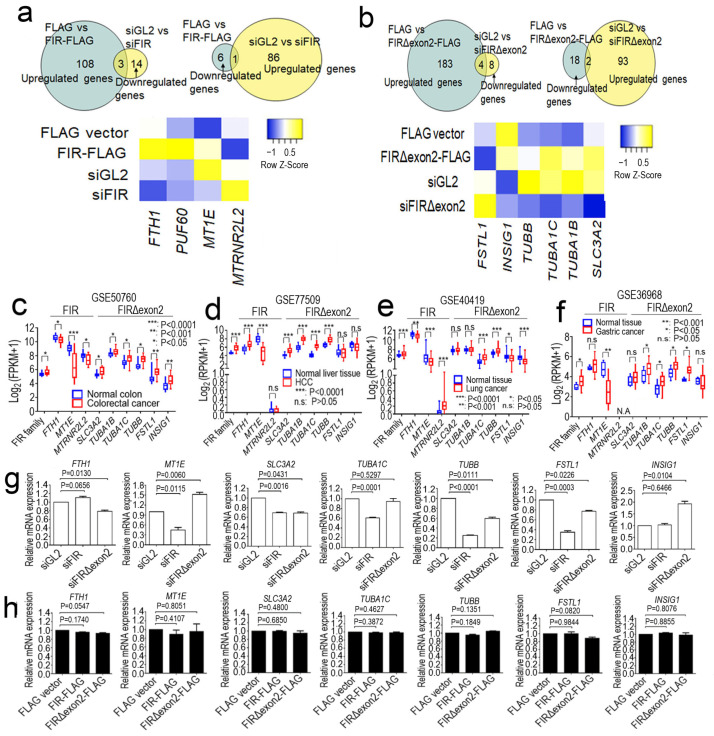
FIR and FIRΔexon2 engaged in the transcription of various cancer−related mRNAs. (**a**,**b**) HeLa cells transfecting with a FLAG vector, FIR−FLAG, and FIRΔexon2−FLAG or siGL2, siFIR, and siFIRΔexon2. Combination analysis of differentially expressed genes (DEGs) under the conditions of overexpression (green) and knockdown (yellow). In FIR, the overlapping DEGs (yellow green) are *FIR/PUF60*, *FTH1*, *MT1E*, and *MTRNR2L2*. In FIRΔexon2, the overlapping DEGs (yellow green) are *SLC3A2*, *TUBA1B*, *TUBA1C*, *TUBB*, *FSTL1*, and *INSIG1*. These DEGs are visualized using heatmaps. Z score presents genes of downregulated (blue) and upregulated (yellow) expression. (**c**) The endogenous gene candidates are up or downregulated in colorectal cancer tissues compared with normal tissues. These candidates are *FIR/PUF60*, *FTH1*, *MT1E*, and *MTRNR2L2* in FIR and *SLC3A2*, *TUBA1B*, *TUBA1C*, *TUBB*, *FSTL1*, and *INSIG1* in FIRΔexon2. (**d**) The endogenous gene candidates are up or downregulated in hepatocellular carcinoma tissues (HCC) compared with normal tissues. (**e**) The endogenous gene candidates are up or downregulated in lung cancer tissues compared with normal tissues. (**f**) The endogenous gene candidates are up or downregulated in gastric cancer tissues compared with normal tissues. (**g**) Relative *FTH1, MT1E*, *SLC3A2*, *TUBA1B*, *TUBA1C*, *TUBB*, *FSTL1*, and *INSIG1* gene expressions under siFIR and siFIRΔexon2 in HCT116 cells. (**h**) Relative *FTH1, MT1E*, *SLC3A2*, *TUBA1B*, *TUBA1C*, *TUBB*, *FSTL1*, and *INSIG1* gene expressions under the overexpression of FIR- and FIRΔexon2-FLAG in HCT116 cells.

**Figure 2 ijms-24-17341-f002:**
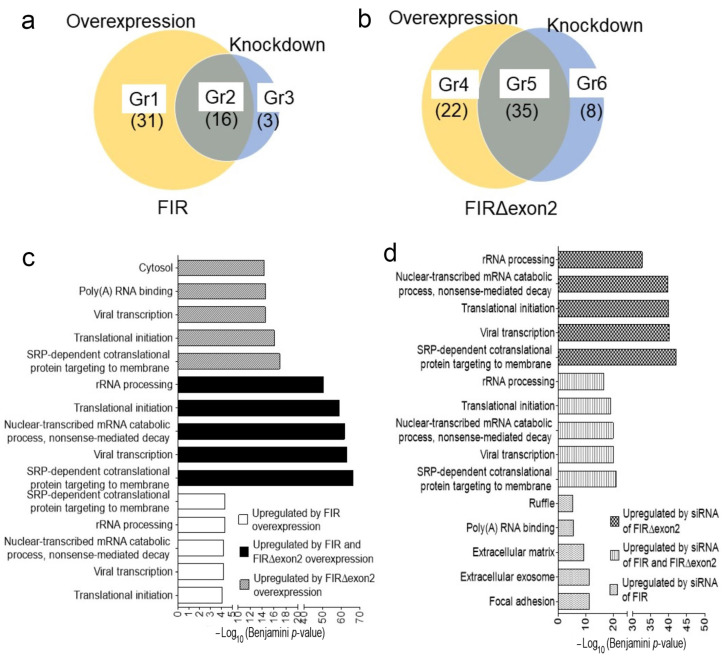
FIR and FIRΔexon2 affected the transcription of various rRNAs. (**a**) Venn diagrams of the RP genes show the overlap (gray:Gr2) of the overexpression (yellow:Gr1) and knockdown (blue:Gr3) in FIR. The RP gene of Gr1, Gr2, and Gr3 are presented (Appendix A). (**b**) Venn diagrams of the RP genes show the overlap (gray:Gr5) of the overexpression (yellow:Gr4) and knockdown (blue:Gr6) in FIRΔexon2. The RP gene of Gr4, Gr5, and Gr6 are presented (Appendix A). (**c**) The bar graph presents the GO term of upregulated genes (FDR < 1) in FIR−FLAG, FIR−FLAG, FIRΔexon2−FLAG, and FIRΔexon2−FLAG. (**d**) The bar graph presents the GO term of upregulated genes (FDR < 0.05) in siFIR and siFIRΔexon2. The GO terms are included with the RP genes. Statistical analyses are conducted using the Benjamini−Hochberg method.

**Figure 3 ijms-24-17341-f003:**
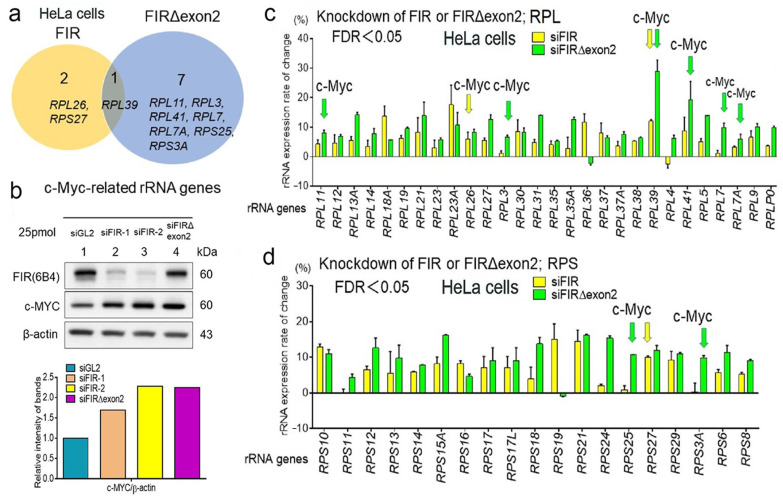
Expressions of the majority of the rRNAs by FIR and FIRΔexon2 were independent of the c-Myc activation. (**a**) c-Myc expression is upregulated by the knockdown of FIR (yellow) or FIRΔexon2 (blue) by siRNA. The Venn diagrams show the overlap (gray) of the c-Myc-related RP genes affected by the knockdown of FIR and FIRΔexon2 by siRNA. The genes affected by both the up- and down-expressions of FIR were *RPL26*, *RPS27*, and *RPL39* (Appendix A (Gr3)). The genes affected by both the up- and down-expressions of FIRΔexon2 are *RPL11*, *RPL3*, *RPL39*, *RPL41*, *RPL7*, *RPL7A*, *RPS25*, and *RPS3A* (Appendix A (Gr6)). (**b**) c-Myc-related rRNA genes detected via knockdown of FIR or FIRΔexon2 by siRNA. HeLa cells are transfected with siGL2, siFIR, and siFIRΔexon2. The c-Myc expression is upregulated by siFIR−1 and siFIR−2 (lanes 2 and 3). Original gels/blots are presented (Appendix A). (**c**) The bar graphs present the RPL rRNA gene expression rate by the knockdown of FIR or FIRΔexon2 (c-Myc was activated in the examined samples, arrows (Appendix A (RPL rRNA genes of Gr3 and Gr6)). (**d**) The bar graphs present the RPS rRNA gene expression rate by the knockdown of FIR or FIRΔexon2 (c-Myc was activated in the examined samples, arrows (Appendix A (RPS rRNA genes of Gr3 and Gr6)).

**Figure 4 ijms-24-17341-f004:**
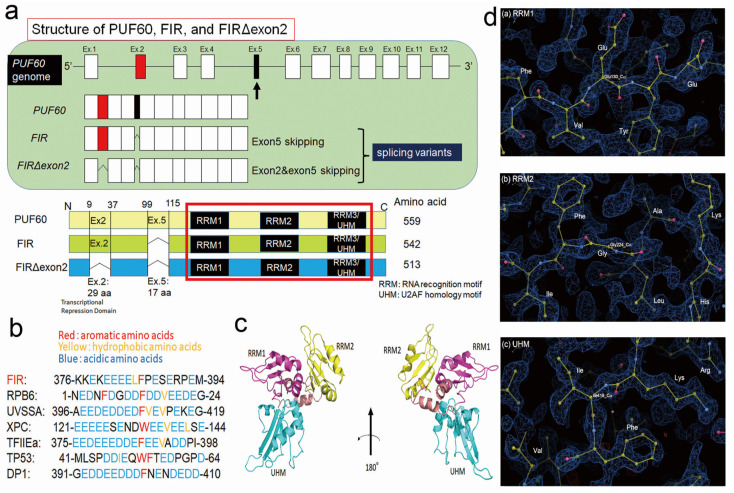
FIR and FIRΔexon2 have highly homologous sequences with RPB6 between RRM2 and RRM3/UHM. (**a**) The structure of PUF60/FIR consists of alternative splicing variants, PUF60, FIR, and FIRΔexon2. (**b**) RNAPI, RNAPII, and RNAPIII contain a common amino-terminal tail (NTT) in RPB6. This NTT of RPB6 interacts with the PH domain of the P62 subunit of TFIIH. These acidic strings of NTT are highly conserved regions among RPB6, UVSSA, XPC, TFIIEα, TP53, and DP1. Furthermore, FIR contains the highly conserved acidic string region (376-KKEKEEEELFPESERPEM-394) between RRM2 and RRM3/UHM. D and E are acidic amino acids targeting the P62-binding site. (**c**) Structure of FIRΔexon2. The crystal structures were previously reported for the three partial domains, namely, RRM1(PDB ID:2QFJ) [42], RPPM2 (PDB ID:2KXH) [16], and UHM (PDB ID:3DXB) [9]. The PDB codes of these partial structures are 2QFJ, 2KXH, and 3DXB. In this study, the recombinant FIRΔexon2 protein is expressed in the form of the whole body. (**d**) The assigning of protein structure to the electron map of X-ray analysis. The blue mesh shows electron density around atoms. Yellow, red, and blue dots represent carbon, oxygen, and nitrogen atoms in the ball and stick representation. Aromatic rings of Phe are clearly visible. The central regions of (**a**) RRM1, (**b**) RRM2, and (**c**) RRM3/UHM are presented in the 2Fo-Fc electron density map with the assigned atom geometry. The residue names are labeled at their side chains. A structural analysis is conducted with the molecular replacement with crystal structures, 2QFJ and 3DXB, as templates. The RRM1, RPPM2, and UHM domains fit well with the electron density map. Accordingly, the X-ray crystal analysis elucidates the relative positions of these three domains.

**Figure 5 ijms-24-17341-f005:**
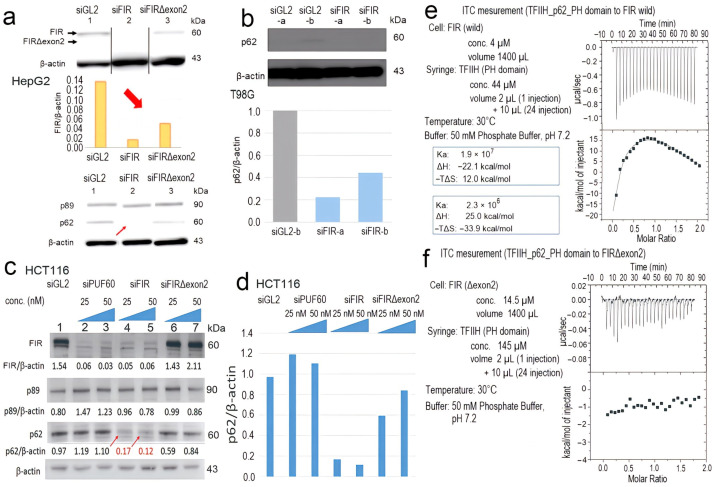
FIR, but not FIRΔexon2, interacts with P62 of TFIIH. (**a**) The siRNA of FIR and FIRΔexon2 significantly suppressed FIR expression (arrow in bar graph) . The siRNA of FIR significantly suppressed P62 expression (arrow) in HepG2 cells. (**b**) The siRNA of FIR significantly suppressed P62 expression in T98G cells. (**c**) Western blot analysis of relative P62 expression via the knockdown of FIR (P62/β−actin = 0.17 and 0.12:arrow) or FIRΔexon2 (P62/β−actin = 0.59 and 0.84) siRNA is evaluated using β−actin as an internal control. Original gels/blots are presented in Appendix A. (**d**) Densitometry analysis of relative P62 expression via the knockdown of FIR or FIRΔexon2 siRNA is evaluated using β−actin as an internal control. (**e**) The ITC measurement of FIR wild with the P62 subunit of TFIIH is suggestive of the molecular interaction between the two proteins. The squre dots in ITC curve indicates the heats released every 10 μL injection of TFIIH to FIR. Exothermic peaks were observed in the initial injections. The peak level decreased in the subsequent injections. The changes in the titration curve indicate that the association between FIR and TFIIH P62 is a two−step sequential process or that the association induces a conformational change of proteins. As the binding reaction is exothermic, the binding of FIR and the P62 subunit of TFIIH is enthalpically driven. This result is compatible with the importance of the highly conserved acidic string region (376−KKEKEEEELFPESERPEM−394) of FIR in molecular binding. (**f**) The ITC revealed the interaction between FIRΔexon2 and the P62 subunit of TFIIH. The binding of P62 of TFIIH to FIRΔexon2 is not significant, whereas that to FIR wild is enthalpically favorable.

**Figure 6 ijms-24-17341-f006:**
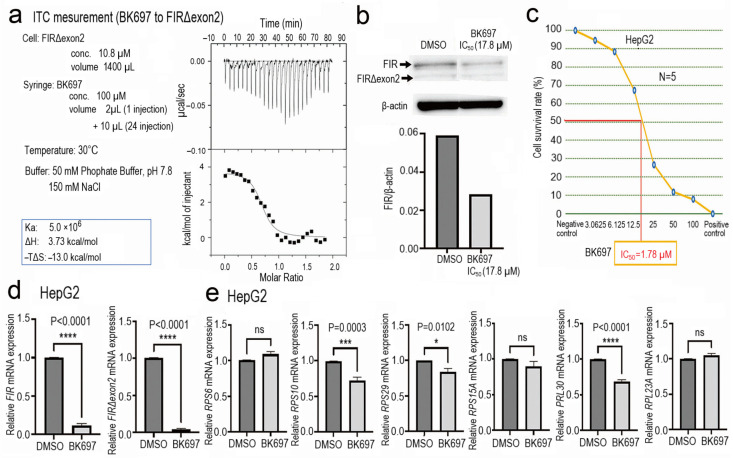
Cell growth inhibition with altered rRNA expression by a low-molecular-weight chemical, BK697, against FIRΔexon2. (**a**) ITC for BK697 and FIR∆exon2 binding. The binding affinity between FIR∆exon2 and BK697 is measured using the MicroCal VP-ITC equipment. The sample cells are filled with 1400 μL of 50 mM phosphate-buffered saline (pH 7.8) containing 150 mM of NaCl and 10 μM of purified FIR∆exon2. The solution containing 100 μM BK697 is injected into the sample cells from the syringe for titration. The total number of injections given is 25, the injection volume is 10 μL each, the injection time is 20 s, and the injection interval is 200 s. The measurement is performed at 30 °C. The squre dots in ITC curve indicates the heats released every 10 μL injection of BK697 to FIRΔexon2. Because the injection of BK697 into the analytical cell generates a large heat of dissolution, the signal for the BK697 binding to FIR∆exon2 is processed by subtracting the influence of the dissolution. (**b**) BK697 suppresses FIR and FIRΔexon2 expressions in HepG2 cells. (**c**) Affiliated chemicals are screened and indicated that BK697 effectively suppresses HepG2 cell growth with an IC_50_ of 17.8 μM. (**d**) BK697 suppresses FIR and FIRΔexon2 mRNA expression in HepG2 cells. (**e**) The expressions of *RPS6*, *RPS10*, *RPS15A*, *RPL30*, and *RPL23A* are reduced by BK697 at an IC_50_ of 17.8 μM in HepG2 cells. *p*-values indicate as follow; ns: *p* > 0.05; * *p* < 0.05; *** *p* < 0.001; **** *p* < 0.0001.

**Figure 7 ijms-24-17341-f007:**
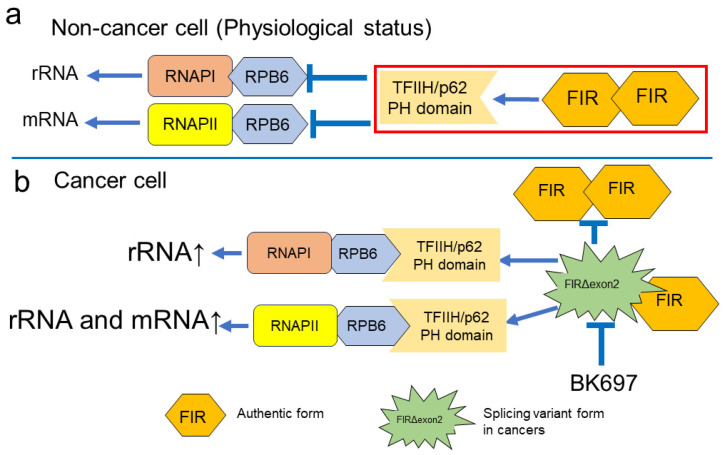
FIRΔexon2 inhibits the FIR–P62 interaction as dominant negative for FIR, causing P62-RPB6/RNAPI/II interaction for rRNA and mRNA activation. (**a**) FIR forms homodimer and heterodimerizes with FIRΔexon2 through RRM1 and RRM2. In non-cancer condition normal cells, FIR interacts with TFIIH/P62, whereas in cancer cells, FIRΔexon2 interferes with the P62–FIR interaction and P62 binds to the RPB6 of RNAPI and RNAPII. (**b**) The existence of FIRΔexon2 due to the disordered RNA splicing drives and accelerates rRNA gene transcription in the nucleolus (arrow). The RNAPI to RNAPII switching by FIRΔexon2 leads to atypical rRNA and mRNA transcription (arrow). BK697 targeted FIRΔexon2 suppression. These results elucidated the novel mechanism indicating that FIRΔexon2 regulates dynamic rRNA expression and is a potential therapeutic target for cancer.

## Data Availability

The RNA-seq data in this study are deposited in the GEO database available as GSE206465 (GSM6254981-GSM6254994). BK697 is available by requesting K.M.

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
