# Peer review of "The Link of mRNA and rRNA Transcription by PUF60/FIR through TFIIH/P62 as a Novel Therapeutic Target for Cancer"

_ijms, 2023, doi:10.3390/ijms242417341_

Round 1
Reviewer 1 Report
Comments and Suggestions for Authors
The MS is well written & describes transcriptional control of selected genes.
However, the Figure 7 gives description comparing normal cells & cancer cells, whereas, the the authors have used only Hela & HCT cancer cells.
How did the authors compare & conclude the findings needs to be explained.
Reviewer 2 Report
Comments and Suggestions for Authors
Kitamura et al., investigates the link between mRNA and rRNA transcription in cancer, focusing on the role of the FUBP1-interacting repressor (FIR) and its variant FIRΔexon2 in influencing these processes through interactions with transcription factor components. It demonstrates that FIRΔexon2 promotes coactivation of mRNA and rRNA transcription, leading to the discovery of a chemical inhibitor, BK697, that targets this pathway to suppress cancer cell growth, highlighting a potential new avenue for cancer treatment research.
This paper contributes valuable insights into the regulation of mRNA and rRNA transcription in cancer cells by FIR and FIRΔexon2, offering potential applications in cancer research and broader therapeutic strategies. While the scientific content is interesting, the presentation of this manuscript is significantly flawed. It requires extensive editing and revision for clarity and coherence before it is suitable for publication.
Major concern:
1. In Figures 1A and 1B, the differentially expressed genes (DEGs) are currently displayed as pie charts. Presenting these DEGs in a heatmap format would offer a more effective visualization, enhancing the clarity and interpretability of the data.
2. The labeling on figures, especially Figures 1, 5, and 6, is too small to be readable and needs enlargement for clarity.
3. The paper should clarify how siFIRΔexon2, a spliced variant of FIR, was selectively knocked down in HepG2 cells while maintaining intact FIR, to provide a more comprehensive understanding of the experimental methodology.
4. The result section is overburdened with excessive discussion, detracting from readability and lacking clear rationales for the experiments conducted. Simplifying this section and including clear rationales for each experiment would enhance clarity and relevance.
5. The discussion section is overly detailed and needs significant refinement to effectively convey the study's findings. It should succinctly summarize key results, place them in the context of existing research, and address both concurrences and deviations from previous studies in a clear and structured manner.
Round 2
Reviewer 2 Report
Comments and Suggestions for Authors
Comments 2: The labeling on figures, especially Figures 1, 5, and 6, is too small to be readable 2 and needs enlargement for clarity.
Still too small to read.
Comments 4: The result section is overburdened with excessive discussion, detracting from readability and lacking clear rationales for the experiments conducted. Simplifying this section and including clear rationales for each experiment would enhance clarity and relevance
Was not addressed properly "The result section is overburdened with excessive discussion"
